# Human Population Density Influences Genetic Diversity of Two *Rattus* Species Worldwide: A Macrogenetic Approach

**DOI:** 10.3390/genes14071442

**Published:** 2023-07-14

**Authors:** Chrystian C. Sosa, Catalina Arenas, Víctor Hugo García-Merchán

**Affiliations:** 1Evolution, Ecology and Conservation Research Group—EECO, Biology Program, Faculty of Basic Sciences and Technologies, Universidad del Quindío, Armenia 630004, Colombia; ccsosaa@javerianacali.edu.co (C.C.S.); catalinaarenascalle@gmail.com (C.A.); 2Department of Natural Sciences and Mathematics, Pontificia Universidad Javeriana, Cali 7 #40-62, Bogotá 110311, Colombia

**Keywords:** invasive species, systematic review, urban landscape genetics, metadata, comparative phylogeography

## Abstract

On a planet experiencing constant human population growth, it is necessary to explore the anthropogenic effects on the genetic diversity of species, and specifically invasive species. Using an analysis that integrates comparative phylogeography, urban landscape genetics, macrogenetics and a systematic review, we explore the worldwide genetic diversity of the human commensal and anthropogenic species *Rattus rattus* and *Rattus norvegicus*. Based on metadata obtained considering 35 selected studies related to observed heterozygosity, measured by nuclear molecular markers (microsatellites, Single Nucleotide Polymorphisms—SNPs-, restrictition site-associated DNA sequencing -RAD-Seq-), socioeconomic and mobility anthropogenic factors were used as predictors of genetic diversity of *R. rattus* and *R. norvegicus*, using the Gini index, principal component analysis and Random Forest Regression as analysis methodology. Population density was on average the best predictor of genetic diversity in the *Rattus* species analyzed, indicating that the species respond in a particular way to the characteristics present in urban environments because of a combination of life history characteristics and human-mediated migration and colonization processes. To create better management and control strategies for these rodents and their associated diseases, it is necessary to fill the existing information gap in urban landscape genetics studies with more metadata repositories, with emphasis on tropical and subtropical regions of the world.

## 1. Introduction

Today, some 56% of the world’s population—4.4 billion inhabitants—live in cities. This trend is expected to continue, with the urban population more than doubling its current size by 2050, at which point nearly 7 of 10 people will live in cities [1]. This growth, especially concentrated in urban and suburban areas of developing countries around the world, subject to anthropic processes such as migration, waste management, pollution and unplanned housing construction, results in landscape modification [2].

The ecological and evolutionary dynamics that occur in the species that inhabit these modified environments can be explored from a genetic perspective to quantify the degree of impact that urbanization and anthropic effects have on the behavior of genetic diversity and population structure [3]. In the case of invasive species, the issue becomes urgent, given the proven consequences of their negative impact on the vertiginous decline of native biodiversity [4]. Given that urban environments are conducive habitats for commensal rodents, compared to more natural ones [5], it is important to assess the genetic diversity of species in environments with some degree of anthropogenic intervention.

The black rat (*Rattus rattus*), native to the India subcontinent, and the brown rat (*Rattus norvegicus*), native to Siberia and China, are two rodent species that have adapted to urban life, spreading to and invading many cities around the world [6]. Among their negative impacts on humanity, the most relevant are the extinction, decrease or displacement of native species [7,8]; transmission of zoonotic diseases [9,10]; and economic losses [11,12].

However, with these negative impacts identified, it is surprising that there are not as many studies exploring the genetic diversity and structure of species with cosmopolitan distribution such as those of the genus *Rattus* mentioned here. It would be expected, according to [6], that both *R. rattus* and *R. norvegicus* would be highly studied species at the genetic level, supported by aspects such as distribution in cities and urban centers around the world, accessible sampling and in large numbers, relatively short generation times, strong ecological associations with the urban habitat and well-developed genomic resources. However, there are very few studies on the subject, particularly in urban environments. For example, in a review by [3], of 470 studies in landscape genetics, only 32 were identified between 2015 and 2020 related to urban landscape genetics, and only 1 was on rodents other than the genus *Rattus*. 

The distribution of genetic diversity of cosmopolitan species mediated by human migrations, such as *R. rattus* and *R. norvegicus*, has a historical component that needs to be considered. Comparative phylogeography offers a framework for understanding such historical factors that have shaped the genetic diversity of species and in turn the connection with contemporary processes explored by landscape genetics [13]. The integration of both approaches may help to unravel the complex interplay between historical events and ongoing landscape processes in shaping the genetic variation of *Rattus* species, and thereby complement the focus of the present study on the distribution of genetic diversity of these two rodent species. In turn, there are two areas that can currently serve as a bridge to identify information to support phylogeographic studies at a landscape scale: namely macrogenetics and systematic reviews. 

Macrogenetics is an emerging discipline that explores the genetic patterns and phenomena associated with large-scale populations by aggregating and reanalyzing thousands of previously published genetic datasets [14]. For systematic reviews, the goal of the field is to provide a valid summary of primary research findings through an explicit, pre-planned procedure [15]. With a growing body of scientific literature in ecology, evolution, genetics, conservation biology and related fields, there is an increasing need to summarize trends, identify emerging questions, clarify controversies and explain conflicting results [16,17].

To assess the impact that anthropogenic processes may have on the distribution of the genetic diversity of *R. rattus* and *R. norvegicus* in urban areas, we tested the hypothesis that economic indicators and socioeconomic disparities may lead to an uneven distribution of genetic diversity between species in different urban areas of the world. If this is confirmed, from the use of metadata, we could establish new predictors that explain the behavior of genetic variability of both species differentially in urban environments. This would in turn have implications for species management and control strategies in cities.

## 2. Materials and Methods

This systematic review was conducted rigorously following the Preferred Reporting Items for Systematic reviews and Meta-Analyses (PRISMA) guidelines [18]. We collected relevant data that conformed to the eligibility criteria of our study. From the systematic review, we carried out a macrogenetic approach, defining predictors of genetic diversity.

### 2.1. Study Design

Based on the absence of a worldwide exploration of the genetic diversity status of two invasive species with cosmopolitan distribution, such as *R. rattus* and *R. norvegicus*, we conducted a systematic review to evaluate at least one metric of genetic diversity of these species according to a series of socio-economic and human mobility predictors.

### 2.2. Eligibility Criteria

#### 2.2.1. Inclusion Criteria

Studies were selected based on the following inclusion criteria: (1) articles considering *R. rattus*, *R. norvegicus* or both species as study species; (2) articles focused on phylogeographic studies; (3) articles with Appendix A or tables with genetic diversity metrics; (4) articles considering nuclear or mitochondrial molecular markers; (5) studies carried out in urban or semi-urban landscapes; (6) articles with geographic coordinates for each data reported; (7) articles published in English.

#### 2.2.2. Exclusion Criteria

The exclusion criteria were as follows: (1) posters, conference papers, abstracts and review papers; and (2) pathogenicity studies associated with viruses and bacteria.

### 2.3. Literature Search and Selection of Articles

We searched the Scopus database for information using the following Boolean combinations: “*Rattus* AND genetic* AND geographic”; “*Rattus* AND genetic* AND phylogeograph*”; “*Rattus* AND phylogenetic”. In the first step, we eliminated duplicate items. In the second step, we reviewed the titles and abstracts based on the inclusion and exclusion criteria. In the third step, we selected studies carried out on *R. rattus* and *R. norvegicus* in urban or semi-urban areas (urban–natural transition). In the fourth step, all selected articles were reviewed for their respective citations using the statistics generated by Google Scholar. As a fifth step and as a final inclusion criterion, we selected articles that had Appendix A or tables with at least one genetic diversity metric that was the same for all studies in both species.

### 2.4. Data Extraction

From reading the full text of all selected articles that met the inclusion criteria, we identified that the only metric of genetic diversity that met the requirement of being present in all studies was observed heterozygosity (*Ho*). Furthermore, given that we found deviations from the Hardy–Weinberg Equilibrium (HWE) in several selected studies [6,19,20,21], *Ho* was precisely the most appropriate metric to use in our study, considering that the expected heterozygosity (*He*) assumes HWE. In addition, only nuclear markers were considered for the analyses to reduce the bias of *Ho* values due to mitochondrial markers’ nature and avoid standardization by marker type as well.

### 2.5. Risk of Bias across Studies

In the second phase of the screening, each of the 139 selected studies were read in full to identify the genetic diversity metrics for further analysis at the macrogenetic level. From this process, the following aspects were identified: (1) there was much heterogeneity in the use of the diversity metrics used; (2) several studies presented global values of genetic diversity, omitting locality-specific values; (3) several studies did not include geographic coordinates in the data analyzed; (4) several studies were focused on the analysis of population genetic structure, presenting genetic variability values at the interpopulation level but not at the intrapopulation level; (5) several studies used data from other studies, which, when reviewed, could not be used due to lack of information (e.g., the data were not available in the literature); and (6) several studies did not use data from other studies, which, when reviewed, could not be used due to lack of information (e.g., the data were not available in the literature). Therefore, to carry out meta-analyses and macrogenetic analyses from a systematic review, we decided to homogenize the data collection in the studies, increase the number of variables to be considered and provide free access to the information through repositories, tables within the text of the articles and the presence of ample and sufficient Appendix A.

Since only observed heterozygosity (*Ho*) was found across most of the studies recovered by PRISMA methodology, we chose to use anthropic factors under the hypothesis of anthropic factors affecting the genetic diversity of both *Rattus* analyzed. Thus, this study tried to take into consideration only raster data with the available data. This means that no pixel was imputed to reduce the description and predictability to areas where complete data were found for all the predictors described below. This contributed to reducing the spatial bias and improving the reproducibility of the results, but also allowed a fair comparison by avoiding creating synthetic data or including noisy information. Nevertheless, the spatial distribution of analysis for species was not considered here, since there is a lack of genetic diversity studies of both species worldwide that did not allow the use of spatial sampling to avoid spatial autocorrelation.

### 2.6. Definition of Predictors

To identify the anthropic factors (economic and human population indicators) affecting the genetic diversity of *R. norvegicus* and *R. rattus* species, a set of five gridded predictors were downloaded: Gross Domestic Product per capita (GDP) for the year 2015 [22]; Population Density, v4.11 for the year 2020 [23]; Global Gridded Relative Deprivation Index (GRDI), v1 for the year 2020 [24]; travel time in minutes to the nearest urban area between 5000 and 110 million people; and travel time in minutes to any port [25]. These predictors were chosen to test whether they were related to perturbations in the genetic diversity of the rat species (*Ho*) from a holistic perspective, according to the following criteria: (1) GDP was chosen to observe if economic growth affects the *Ho* values of the two species; (2) human population density was chosen to observe if the human population density is related to higher *Ho* values; (3) the relative levels of multidimensional deprivation and poverty affecting rats genetic diversity were chosen to observe how the closeness to main cities or port affects the *Ho* value of both rat species. All datasets were resampled to 5 km (2.5 arcmins) using the nearest-neighbor method to reduce the computation time and standardize data to the same resolution for the description and prediction of observed heterozygosity via random forest regression.

Predictor values were extracted for each of the localities extracted from the systematic search using the Raster v3.6-20 R package. Localities were omitted from the analysis if they did not possess values for all the predictors for each species (localities with no values were discarded from further analyses). A principal component analysis (PCA) was performed using FactoMineR v.7 to inspect visual similarities between *R. norvegicus* and *R. rattus* localities sampled according to observed heterozygosity and the five predictors described above. 

PCAs were performed for each species to detect predictors of multicollinearity and perform a hierarchical principal component clustering (HCPC) with the FactoMineR v.7 R package to detect groups of localities per species. On the other hand, random forest regressions were performed to detect the impact of the predictors on the observed heterozygosity. A Jackknife validation strategy (without sample replacement), given the small number of samples per species (<100), was used. Likewise, as a criterion of importance to predict the effect on observed heterozygosity, we used the Gini index. In addition to the Jackknife validation strategy, Wilcoxon tests were performed for *Ho* to identify the influence of mainland and island effect over the presence of both rat species. No regression predictions were performed for island and mainland records separately, since the number of island and mainland records were low for both species: *Rattus norvergicus* (island = 24 records, mainland = 59 records), and *R. rattus* (island = 24 records, mainlaind = 35 records). 

A gridded search to find the random forest number of trees from a range from 10 to 10,000 trees was performed. The optimal random forest regression configuration per species was defined as the regression with the minimum value of out-of-bagging value (OOB) root mean square error (RMSE), and it was used for all the subsequent analyses. These analyses were performed in the ranger R package v0.14.1.

A worldwide prediction of observed heterozygosity was obtained for each *Rattus* species using the random forest regression and the spatial predictors. To obtain an evaluation of possible extrapolation and uncertainty for the predictors used to project the predicted observed heterozygosity spatially, a Multivariate Environmental Similarity Surface (MESS) was calculated with the spatial predictors GDP, Population Density, GRDI, travel time in minutes to the nearest urban area between 5000 and 110 million people and travel time in minutes to any port, extracted from the localities obtained to train the random forest regressions for each species. This approach was preferred, given that the MESS index highlights areas where environmental conditions are outside conditions found in the presence records [26]. Negative MESS values indicate areas where a projection of a model based on the predictors described above would be due more to extrapolation than a proper prediction. In consequence, to avoid uncertainty in the projected *Ho* values, only observed heterozygosity grids greater than the median of the MESS positive values were considered valid predicted grids to reduce the bias of extreme predicted values in maps. MESS maps are available in Appendix A.

## 3. Results

### 3.1. Systematic Review

A total of 1032 articles were identified using our search strategy, as shown in Figure 1. Ultimately, 35 studies (17 for *R. rattus* and 18 for *R. norvegicus*) were systematically reviewed (Appendix A).

In terms of geographic distribution, 8 studies considered data from Africa, 9 from the Americas, 12 from Asia and 10 from Europe (Figure 2). Twenty studies used microsatellite molecular markers, ten used SNPs and seven used genomic approaches. At the level of studies comparing molecular vs. mitochondrial markers, eight were identified. Three studies carried out a comparative phylogeography approach, two considering *R. rattus* and *R. norvegicus* and one a different species of the genus. Only one study combined a comparative phylogeography approach considering both nuclear and mitochondrial molecular markers. As for the fine spatial scale, 8 studies evaluated anthropogenic processes affecting genetic diversity in cities and 10 explored it in archipelagos. We found 12 studies where the focus was associated with migration routes and historical colonization processes of *R. rattus* and *R. norvegicus*.

### 3.2. Metrics of Genetic Diversity

In the 35 studies considered, genetic diversity measured as observed heterozygosity was higher for *R. rattus* (0.623 ± 0.126) than for *R. norvegicus* (0.453 ± 0.181). For *R. rattus*, the highest values were recorded in Southern India and the lowest in Bowditches, United States (Appendix A). For *R. norvegicus*, the highest value was found in West Baltimore, United States, and the lowest in Helsinki, Finland (Appendix A).

A low correlation was observed between population density and observed heterozygosity for *R. norvegicus.* In contrast, three predictors, gridded poverty (deprivation value), travel time to cities (min) and travel time to ports (min), showed light to medium correlations for *R. rattus*, but a slight effect could be observed with the travel time to ports for the last species. Thus, none of the predictors for both species had multicollinearity with the *Ho* values (Table 1)

The predicted values of *Ho* using MESS did not allow prediction values for Northern America due to the presence of negative values, which indicates a possible extrapolation that was omitted to ensure the reproducibility of our approach. Nevertheless, the highest predicted *Ho* values were found in Western Europe, Japan, the mountain region of Latin-America and southwestern India for *R. rattus* (Figure 3). In contrast, for *R. norvegicus*, the highest values of predicted *Ho* were in Northern Africa, South India, Indochina, Eastern Europe and South America. Finally, Western countries showed individuals with lower predicted *Ho* values (Figure 4).

### 3.3. Predictors of Genetic Diversity

Random forest regression performance metrics were highly similar between observed and predicted *Ho* values, as is suggested by the root mean square error (RMSE) of the out-of-bag evaluation, RMSE = 0.0229 and RMSE = 0.005 for *R. norvergicus* and *R. rattus*, respectively. A similar behavior was observed for the RMSE of observed vs. predicted *Ho* values, RMSE = 0.095 and RMSE = 0.0422 for *R. norvergicus* and *R. rattus*, respectively. This indicates that both regressions accurately predicted the training and validation dataset values used in the Jackknife strategy (Figure 5)

The Gini index values using random forest regression methodology (Table 2) and principal component analysis (Figure 6) supported our working hypothesis, indicating that on average, human population density is the best predictor of genetic diversity (measured as observed heterozygosity) for both *R. rattus* and *R. norvegicus*. Nevertheless, it is important to highlight that *R. rattus* island records had more variability than the mainland records (Wilcoxon tests, *p*-value 0.0038), but this does not affect our original working hypothesis. 

Other predictors identified by the analyses in the characterization of genetic diversity were travel time to ports and Gross Domestic Product (GDP) per capita for *R. rattus* and the global Gridded Relative Deprivation Index (GRDI) and GDP per capita for *R. norvegicus*.

## 4. Discussion

This study is the first, to our knowledge, using metadata obtained from a systematic review combined with a comparative phylogeographic and landscape genetic analysis, to evaluate urban anthropogenic effects from a genetic perspective in two rat species widely distributed around the world. According to [14,27], our exploration may be an application of the new area known as macrogenetics. From Gini’s index using random forest regression and the results of principal component analysis, we found that, on average, population density is the main predictor determining genetic diversity (measured by observed heterozygosity in nuclear markers) in *R. rattus* and *R. norvegicus* (Table 2 and Figure 6). Considering all metadata used for both species, the observed heterozygosity was higher in *R. rattus* than in *R. norvegicus*, which is consistent with the distribution of both species in the United States [28]. We confirm the hypothesis that anthropogenic factors, such as population density and GDP per capita, are predictors of differential behavior of genetic diversity in *R. rattus* and *R. norvegicus* through metadata obtained from 35 studies analyzed worldwide.

The effects of anthropogenic factors on the genetic diversity of invasive species in urban habitats can be complex and depend on many aspects associated with human activities. In our study, we focused on socioeconomic and mobility indicators as predictors of genetic diversity. The effects of anthropogenic factors on the genetic diversity of invasive species in urban habitats can be complex and depend on many aspects associated with human activities. The signal predictors used here can be interpreted in the light of five interrelated points: (1) species dispersal; (2) heterogeneity and scale of urban environments; (3) migration and colonization processes of invasive species; (4) species behavior defined by life history traits; and (5) availability of metadata-type information.

Several studies evaluating the genetic variability of species of the genus *Rattus* indicate that dispersal is low or very local, at the scale of cities and even neighborhoods within the same city, indicating the effect that the urban environment may have on the distribution of genetic diversity [3,6,19,20,21]. For example, in the systematic review conducted herein, we found several studies [19,20,21] where an absence of correlation between genetic and geographic distances was presented, which can be expected when passive (e.g., human-mediated) dispersal or heterogeneity occurs on a fine scale in the urban habitat. However, in our case, by exploring other types of correlations (e.g., population density vs. *Ho*; gridded poverty/deprivation value vs. *Ho*; travel time to cities/ports vs. *Ho*), we were able to identify some form of relationship (Table 1). This indicates the potential for exploring variables that directly include genetic diversity (measured by heterozygosity) and predictors that transcend the classical model of isolation by distance. 

Likewise, the genetic patterns of *R. rattus* and *R. norvegicus* will depend on the interactions between demographic and behavioral processes, on the one hand, and the spatial scale at which each site has been sampled on the other. Thus, the genetic diversity of these invasive species is defined by the interactions between their dispersal capacity, the heterogeneity and quality of habitats (in terms of food availability) and the social behavioral characteristics per se of the species, such as philopatry and their level of territoriality and aggressiveness [21,29,30,31].

High population density can also create barriers to gene flow and dispersal, which can reduce the genetic diversity of invasive species populations. For example, urban areas can be surrounded by barriers such as highways, rivers or mountains, which can restrict the movement of invasive species and lead to genetic isolation. In our study, population density was the main predictor identified for *R. norvegicus* (Gini index = 1) and the second for *R. rattus* (Gini index = 0.74). In this regard, there is no study that has systematically addressed this predictor in the distribution of genetic variability in invasive rodents. The only study that tangentially approached it was [32], which explored globally the effect of population density and land use on genetic diversity in more than 150,000 data points of birds, insects, fish and mammals, finding no effect of the two predictors. However, with respect to our study, the inference found in [32] should be analyzed with caution for three reasons: (1) it is based on mitochondrial DNA; (2) it considers only 40 *R. rattus* data points; and (3) most of the points correspond to the same localities (same georeferenced points).

*R. rattus* and *R. norvegicus* are arguably the most successful invasive species on the planet, and human migrations have had a major influence on the spread and dispersal of invasive species populations within and between urban areas through processes such as genetic drift and gene flow [33,34]. Human movements and transportation networks can facilitate the dispersal of rats across urban environments, which can increase gene flow between populations and impact the genetic diversity of populations at different scales [31,35,36,37]. For example, while [19], using SNPs, found divergent genomic signals in *R. norvegicus* in New York City at a scale of less than 600 m between colonies, Ref. [38] found convergence in genetic diversity signal with the same type of molecular marker among *R. norvegicus* individuals from eastern North America, South America, Africa, Australasia and western Europe. 

Moreover, due to the genetic variability detected between different inheritance types of molecular markers in the same study, such as [28,38,39] in our systematic review, it is necessary to define the same type of molecular marker (nuclear, in the meta-analysis of the present study) in order not to confound possible signals about the association between genetic diversity, anthropogenic traits and differentiation of spatial scales of sampling (e.g., islands vs. continental areas). In another study by [31], the importance of river transport and subsequent connectivity between ports is highlighted, which is consistent with travel time to port, one of the predictors of genetic diversity identified by us in *R. rattus*. The history of *Rattus* colonization in the world is the history of human migrations; consequently, the origin of commensalism in these rodents inevitably reflects the history of human settlement and associated anthropogenic factors. Although further studies are needed, the more recent dispersal of *R. norvegicus* from its center of origin in Siberia to Europe in the 16th century [40] may be one of the reasons for the lower average value of genetic diversity found in our metadata, compared to the dispersal of *R. rattus*, which is linked to the beginning of Indian civilization (approximately 5000 years BC) and subsequent dispersal dynamics, mainly by maritime routes [35]. Other evidence supporting this behavior of a higher average value of genetic diversity in *R. rattus* than in *R. norvegicus* was found in two studies with a comparative phylogeography approach to that reported in our study. The first corresponds to a regional scale across the United States [37], and the second to a local scale on a Canadian Pacific Island [41].

## 5. Conclusions and Recommendations

While high human population density can create conditions that favor the establishment and spread of invasive species, it can also create barriers to gene flow and dispersal and increase selective pressures that reduce genetic diversity. Understanding these factors is crucial to managing and mitigating the impact of invasive species in urban areas. In the case of *R. rattus* and *R. norvegicus*, human commensal and anthrodependent species, it is vital to explore the spatiotemporal dynamics of genetic diversity in current and emerging urban areas around the world (with special emphasis on tropical countries) to design management and control strategies for these pests and their associated diseases. For instance, it would be interesting to observe if the changes in *Ho* values are related to changes in the human population to provide tracking systems of which strategies can influence and lead to the mitigation of the dispersal of these invasive species worldwide.

Although we identified several studies in regions other than developed countries and with tropical or subtropical distribution, e.g., Nigeria [31], Benin [42], Senegal [36], Gabon [30]; Brazil [6,21], India [35,43], Madagascar [29] and Ecuador [44], further sampling and genetic diversity analysis efforts are needed in the regions of Asia, Africa and Central and South America, given the specific projections of population expansion expected in the near future in these areas and to the socioeconomic, political, climatic and urban development characteristics, which may be very different from those of temperate zones, where the largest number of studies on landscape genetics of invasive rodent species to date have been carried out.

Our analyses reveal critical data and theory gaps and call for increased efforts to monitor global genetic diversity. To increase the data matrices of invasive species in the world, and thus strengthen management strategies based on empirical evidence, it is necessary to expand the focus on urban landscape genetics in species of the genus *Rattus*, both those covered in this study and others not considered. Following the study by Fusco et al. 2021 mentioned in the introduction, we conducted a search between 2021 and 2023 with the keywords “urban landscape genetics”, considering its appearance anywhere in the body of a paper, finding only 20 studies, of which none were conducted in rodents. This shows that the information gap persists today and needs to be filled.

Human activities can also affect intraspecific genetic diversity at the natural level (e.g., via habitat fragmentation, [45,46]), hence the need for further studies comparing disturbance processes and temporal and spatial modification of evolutionary mechanisms such as selection (e.g., adaptive processes), genetic drift and gene flow (e.g., levels of connectivity) in invasive species distributed in both urban and natural habitats.

To compare the genetic diversity of invasive species at different spatial and temporal scales, it is necessary to strengthen the metadata related to nuclear molecular markers (e.g., [47]) to design macrogenetic studies that assess variability at different temporal scales and spatial data, and thus add to the studies based on larger existing repositories at the mitochondrial DNA level. Likewise, based on its genome distribution, genotyping facilities, and lower homoplasy compared to microsatellites, it is necessary to consider more studies in urban landscape genetics in invasive species with nuclear markers, such as SNPs and genome-wide studies, to improve possible tracking systems of genetic diversity caused by human activities. This integrative work shows the importance of increasing the sampling effort between continental and insular areas, in order to define the criteria for management and control of invasive species with a wide distribution in the world.

*R. rattus* and *R. norvegicus* are considered among the most important invasive species in the world for reasons such as their cosmopolitan distribution, high reproductive capacity, adaptive capacity, negative impact on ecosystems, disease vectors and economic impacts. In this scenario, control strategies are a challenge. Macrogenetic-scale studies such as the one presented here can provide the following recommendations to mitigate the negative effects of these species on the invaded ecosystems: 

(1) By analyzing the genetic composition of rat populations in different areas using metadata, scientists can determine the origin of introductions, colonization patterns and the degree of genetic mixing between populations more precisely. This information could help understand how these species spread and establish new populations.

(2) Macrogenetic studies can help identify the routes and pathways by which rats are introduced into new regions. Through analysis of the genetic signatures of rat populations, the origins and possible sources of invasion can be traced. This knowledge is crucial for implementing targeted prevention measures and biosecurity protocols at points of entry to reduce the risk of new introductions.

(3) Examination of large-scale genetic variation can identify patterns in genetic traits or markers associated with traits such as reproductive capacity, survival, behavior and resistance to control measures. This information can guide the development of more effective control strategies that target specific traits or exploit vulnerabilities in invasive populations.

(4) Different rat populations throughout the world may vary in their behavior, their response to particular control methods or their susceptibility to specific toxins. By accounting for genetic diversity on a global scale, control strategies can be tailored to the unique characteristics of specific populations, increasing the chances of success and minimizing unintended consequences.

(5) Large-scale genetic diversity studies can provide a more robust basis for monitoring the effectiveness of control strategies over time. By comparing the genetic composition of rat populations before and after control efforts, it is feasible to assess the impact of interventions, detect any changes in population structure or genetic diversity and make necessary adjustments to control strategies.

## Figures and Tables

**Figure 1 genes-14-01442-f001:**
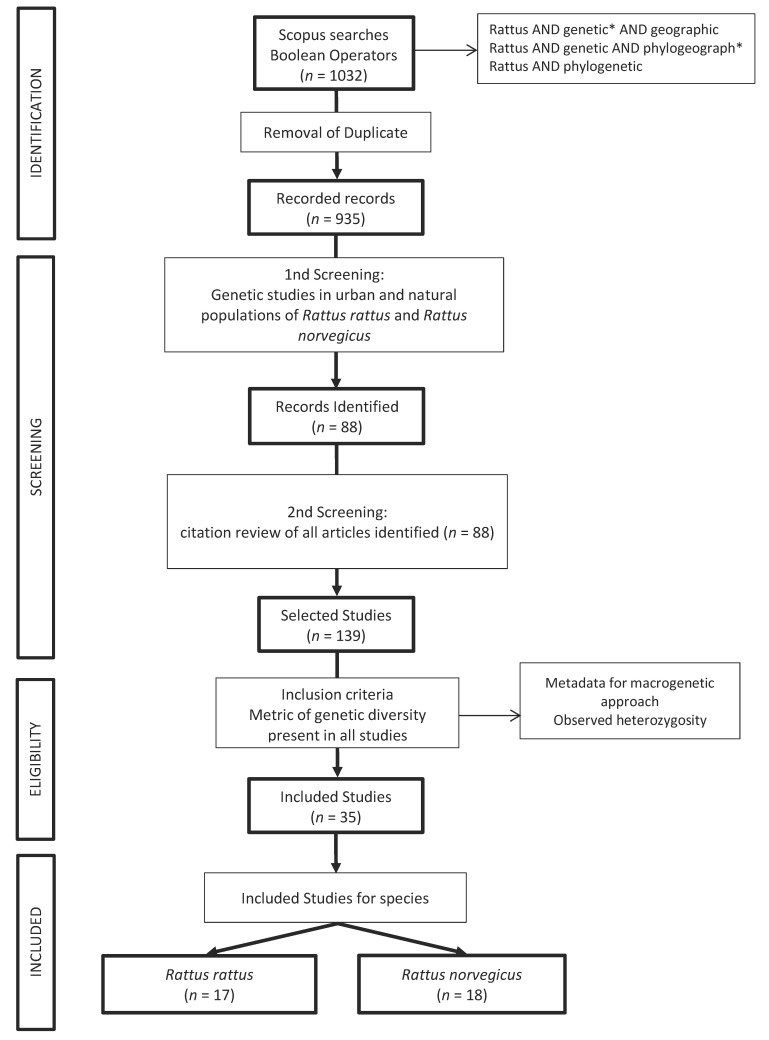
PRISMA flow diagram of literature search and selection for articles included in the systematic review. The asterisk (*) was used to extend the Boolean search to include various word forms and endings (e.g., “phylogeography”, “phylogeographical”). Adapted from [18].

**Figure 2 genes-14-01442-f002:**
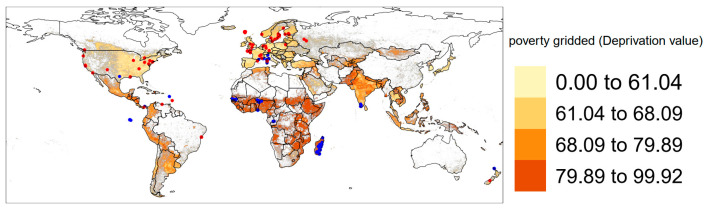
Map with the distribution of sampling points used in the 35 studies considered. Red dots: 141 records of *Rattus rattus*; blue dots: 234 records of *R. norvegicus*. Background: the deprivation index displayed as poverty proxy.

**Figure 3 genes-14-01442-f003:**
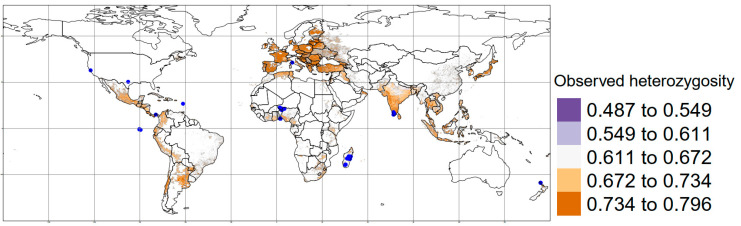
Map with the distribution of observed heterozygosity for *Rattus rattus*, considering the information of 35 studies analyzed. Blue dots: Sampling areas considered.

**Figure 4 genes-14-01442-f004:**
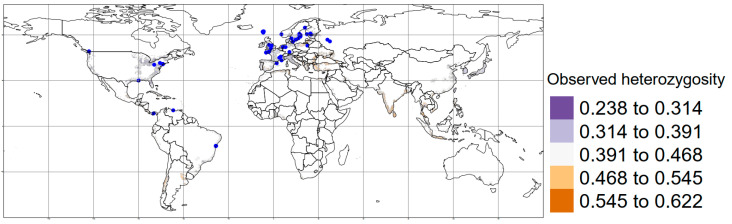
Map with the distribution of observed heterozygosity for *Rattus norvegicus*, considering the information of 35 studies analyzed. Blue dots: Sampling areas considered.

**Figure 5 genes-14-01442-f005:**
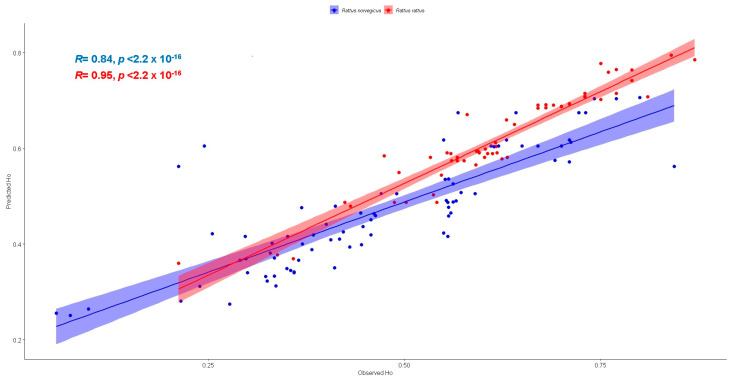
Random forest regression between observed vs. predicted heterozygosity values including regression lines, R² and *p* values per species. *R. norvegicus* (blue); *R. rattus* (red).

**Figure 6 genes-14-01442-f006:**
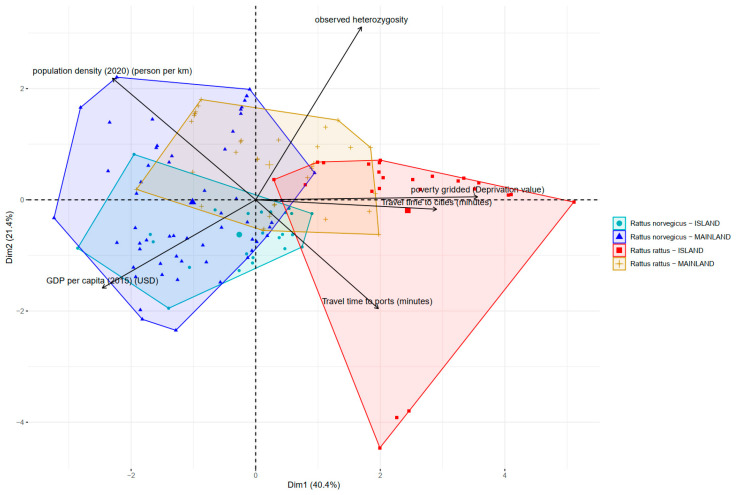
Principal component analysis of predictors of genetic diversity for *R. rattus* and *R. norvegicus*, considering the information of 35 studies analyzed. *R. norvegicus* island records (light blue). *R. norvegicus* mainland records (blue). *R. rattus* island records (red). *R. rattus* mainland records (light brown). The acronyms of the predictors are as described in Table 1.

**Table 1 genes-14-01442-t001:** Pearson correlation values found for the five predictors used in the random forest regression for *Rattus norvegicus’* and *Rattus rattus’* observed heterozygosity. Values in parentheses are the *p*-values of the correlation tests.

Predictor	*R. norvegicus*	*R. rattus*
GDP_per_capita_2015	−0.064 (0.567)	−0.108 (0.417)
gpw_v4_population_density_2020	0.385 (0)	−0.236 (0.072)
povmap.grdi.v1	0.162 (0.143)	0.3 (0.021)
travel_time_to_cities_12_MOD	−0.004 (0.97)	0.286 (0.028)
travel_time_to_ports_5_MOD	−0.058 (0.604)	−0.588 (0)

GDP_per_capita_2015 = Gross Domestic Product per capita for the year 2015; gpw_v4_population_density_2020 = Population Density, v4.11 for the year 2020; povmap.grdi.v1 = Global Gridded Relative Deprivation Index (GRDI), v1 for the year 2020; travel_time_to_cities_12_MOD = travel time in minutes to the nearest urban area; travel_time_to_ports_5_MOD = travel time in minutes to ports.

**Table 2 genes-14-01442-t002:** Gini index values obtained for *R. norvegicus* and *R. rattus* using random forest regression.

Predictor	*R. norvegicus*	*R. rattus*
GDP_per_capita_2015	0.743	0.697
gpw_v4_population_density_2020	1	0.741
povmap.grdi.v1	0.720	0.363
travel_time_to_cities_12_MOD	0.205	0.595
travel_time_to_ports_5_MOD	0.565	1

The acronyms of the predictors are as described in Table 1.

## Data Availability

Data, R scripts and results are publicly available in the GitHub repository https://github.com/ccsosa/Rattus_urban_macrogenetics.

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
