# Peer review of "Human Population Density Influences Genetic Diversity of Two Rattus Species Worldwide: A Macrogenetic Approach"

_genes, 2023, doi:10.3390/genes14071442_

Round 1
Reviewer 1 Report
Overall the idea to use published data from around the world too look at the relationship between human populations (density, inequality, connectivity) and commensal rat genetic diversity is interesting and relevant. The genetic data collected from the literature in this case includes different markers that have different characters, including heterozygosity. The statistics do not take into account marker type, and this is important as one would expect the different markers utilized here to have different levels of heterozygosity in the same animals and populations. Another factor that could potentially have a large impact on the heterozygosity that was not incorporated into the statistical analysis is the island vs. mainland distribution of the population. This aspect was also confounded in the Introduction. So, the introduction needs to be clarified and the issues of island (associated with the mentioned biodiversity issues) vs mainland cities needs to be treated separately from human population density. In the statistical analyses it is crucial to make sure the factors of marker type and island/mainland are not confounded with the target questions about the association between genetic diversity and human population characters.
Minor editorial comments:
Lines 137-141- this should be in the discussion not Materials and Methods
Fig 1: how did the number increase between steps during screening?
Fig 2: It would be nice to also see human density or inequality mapped on this map
Fig 3 & 4: It would be nice to see the sampling points, as in Fig S2
Overall the English is fine. It could be easier to read if sentence lengths were reduced, especially in the introduction.
Author Response
Overall the idea to use published data from around the world too look at the relationship between human populations (density, inequality, connectivity) and commensal rat genetic diversity is interesting and relevant.
Thank you very much for this comment. We believe that there is great potential in the data available worldwide that can be used and optimized under new analysis and approaches. Our study aims to draw attention to the importance of unified information in open access repositories to generate new study hypotheses.
The genetic data collected from the literature in this case includes different markers that have different characters, including heterozygosity. The statistics do not take into account marker type, and this is important as one would expect the different markers utilized here to have different levels of heterozygosity in the same animals and populations.
Only nuclear markers were taken into consideration for the analyses to reduce the bias in the analyses. This was clarified in the methods section to avoid confusion for the readers.
Another factor that could potentially have a large impact on the heterozygosity that was not incorporated into the statistical analysis is the island vs. mainland distribution of the population. This aspect was also confounded in the Introduction.
Thank you very much for this comment. We agree with the reviewer that the island and mainland can introduce bias to the analyses. Nevertheless, we performed a descriptive analysis of the predictors and the observed heterozygosity for the localities obtained which are attached here (Figures 1 to 4 of this letter), and only one predictor: travel time to ports presented a strong correlation for Rattus rattus (See figure 4 of this rebuttal letter) with three extreme values affecting the correlation. Also, it is important to highlight that 24 records were available for islands for each of the species which certainly represents a low number of records to split the analysis between mainland and island records. Moreover, the Random forest regression performs well with non-linear relationships and this was observed with low values of RMSE between observed and predicted values for both species (This information was added to the results section) to gain readability for readers.
So, the introduction needs to be clarified and the issues of island (associated with the mentioned biodiversity issues) vs mainland cities needs to be treated separately from human population density.
In accordance with the importance of the subject mentioned by the reviewer in the previous point, the low number of points plus the use of an algorithm that is able to obtain insights from non-linear relationships justify the fact of avoid of trying island and mainland separately. However, Wilcoxon tests (with only significant value obtained for Rattus rattus for observed heterozygocity), as well as the inclusion of mainland and island categories were included in Figure 6 to use the reviewer comment to improve the manuscript.
In the statistical analyses it is crucial to make sure the factors of marker type and island/mainland are not confounded with the target questions about the association between genetic diversity and human population characters.
A line clarifying that only nuclear markers were used was added to avoid confusion for readers. Also as we explained above, the low number of records (<100) per each species was a strong motivation to explore regressions or even to try separately the marker effect througout the analysis.
Minor editorial comments:
Lines 137-141- this should be in the discussion not Materials and Methods
Thank you very much. This line was included in material and methods to provide our justification to use our approach.
Fig 1: how did the number increase between steps during screening?
Thank you for this observation of an aspect that is not common in conventional systematic reviews. In our case, in the second filter it went from 88 to 139 because we wanted to check that key articles were not included in the citations due to criteria such as access to data.
Fig 2: It would be nice to also see human density or inequality mapped on this map
Thank you for this recommendation that enriches and contrasts better the message we want to give. Poverty gridded (Deprivation index was added to the figure 2 as the reviewer suggested).
Fig 3 & 4: It would be nice to see the sampling points, as in Fig S2
Thank you for this observation. As in the previous case, it improves the magnitude of the message. The points have been included in figures 3 and 4.
Overall the English is fine. It could be easier to read if sentence lengths were reduced, especially in the introduction.
Thank you for this recommendation. We hope that without having shortened the sentences, the document is still to your liking, it was a challenge to build it.
Thank you very much for all the recommendations made to the document and the time taken to make them. We believe that the document has been much better with your valuable contributions. We hope you enjoy it.
P.S.: In the attached pdf file we include the figures that support our answers. Thank you very much for reviewing them

Reviewer 2 Report
Overall, I really enjoyed this paper. It is interesting to see meta-genomic analyses come through for urbanized organisms. The authors conducted a meta-analysis on the genetic diversity of 2 rat species across their native and introduced ranges, based on the available data. I have a few minor comments.
The reasons behind the predictor variables are not clearly stated. Why use GDP? I understand that this is a global dataset, but there is significant heterogeneity in economic factors at multiple smaller spatial scales. It states in line 160-161 that these were samples at 5km, but GDP is at a higher scale than that. It is unclear if the population density was at the 5km point as well. Please clarify.
For the MESS index (lines 182-190), it is unclear what spatial predictors were used. Please clarify (can provide more detail in Supplement if needed). Based on the Supp Fig 1, The majority of the world is at a value of 0, so it is unclear what this provides for the current study. The regions with MESS values different from 0 are where rats were not sampled. Thus it is unclear how this would relate to observed heterozygosity.
I liked that the authors included Fig 1 since this figure helps to conceptualize the steps in the meta-analysis.
A minor point in the results, please indicate whether the correlations are positive or negative. For example, travel time to a port, I would assume that less travel time would increase genetic diversity since they are closer to the port which is the point of entry.
The authors do not state the native range until the last paragraph of the discussion (lines 326-334). Given that the prediction is generally that genetic diversity is higher in native range, this should be brought up much sooner, like in the intro or methods. This would guide the hypothesis that anthropogenic activity may influence genetic diversity.
Throughout the discussion, the authors refer to what other studies have found, but do not clearly state how these are similar or different to what they have found.
The conclusions and recommendations are general to meta-analyses. Given that this is an analysis on an invasive pest, I expected recommendations about what tracking genetic diversity can do to mitigate the negative effects of these species on the invaded ecosystems. Simply identifying patterns of genetic diversity does not inform us on what we can do to eradicate them. There needs to be some link to this. This is not to say that I disagree with their conclusion, there absolutely needs to be better practices of data availability and some standard of data so that these types of meta-analyses can be conducted. I just think that some link as to why they conducted this specific analysis (invasive rate genetics) and what it can provide us would be helpful to include.
Author Response
Overall, I really enjoyed this paper. It is interesting to see meta-genomic analyses come through for urbanized organisms. The authors conducted a meta-analysis on the genetic diversity of 2 rat species across their native and introduced ranges, based on the available data. I have a few minor comments.
The reasons behind the predictor variables are not clearly stated. Why use GDP? I understand that this is a global dataset, but there is significant heterogeneity in economic factors at multiple smaller spatial scales. It states in line 160-161 that these were samples at 5km, but GDP is at a higher scale than that. It is unclear if the population density was at the 5km point as well. Please clarify.
Thank you very much for this recommendation. A paragraph explaining the reason to use the predictors was included into the matherial section.
For the MESS index (lines 182-190), it is unclear what spatial predictors were used. Please clarify (can provide more detail in Supplement if needed). Based on the Supp Fig 1, The majority of the world is at a value of 0, so it is unclear what this provides for the current study. The regions with MESS values different from 0 are where rats were not sampled. Thus it is unclear how this would relate to observed heterozygosity.
Thanks for your gentle comment. A more detailed explanation was included to explain the use of MESS to avoid the extrapolation for the random forest regression in the maps. In addition, only the overlapped areas of high MESS values and predicted Ho values were used to reduce the bias prediction at omit high probable extrapolation areas according to the MESS index.
I liked that the authors included Fig 1 since this figure helps to conceptualize the steps in the meta-analysis.
A minor point in the results, please indicate whether the correlations are positive or negative. For example, travel time to a port, I would assume that less travel time would increase genetic diversity since they are closer to the port which is the point of entry.
Overall correlations were added in order to cover this issue, please see the results secion.
The authors do not state the native range until the last paragraph of the discussion (lines 326-334). Given that the prediction is generally that genetic diversity is higher in native range, this should be brought up much sooner, like in the intro or methods. This would guide the hypothesis that anthropogenic activity may influence genetic diversity.
Thanks so much for this comment. The reported native areas of Rattus rattus and Rattus norvegicus distribution were added into the introduction
Throughout the discussion, the authors refer to what other studies have found, but do not clearly state how these are similar or different to what they have found.
Thank you very much for this observation. We agree that we missed that link between studies. In the new version of the discussion we tried to better link the ideas between what we found and the existing literature.
The conclusions and recommendations are general to meta-analyses. Given that this is an analysis on an invasive pest, I expected recommendations about what tracking genetic diversity can do to mitigate the negative effects of these species on the invaded ecosystems.
Simply identifying patterns of genetic diversity does not inform us on what we can do to eradicate them.
There needs to be some link to this. This is not to say that I disagree with their conclusion, there absolutely needs to be better practices of data availability and some standard of data so that these types of meta-analyses can be conducted. I just think that some link as to why they conducted this specific analysis (invasive rate genetics) and what it can provide us would be helpful to include.
We fully agree with this observation. A study such as the present one, without the considerations of management and control of these invasive species, is just another descriptive study of a problem that is over diagnosed. Thank you very much for this recommendation. We hope that the last paragraph is in accordance with what you have in mind and it is necessary to include.
Thank you very much for all the recommendations made to the document and the time taken to make them. We believe that the document has been much better with your valuable contributions. We hope you enjoy it.